

# Complete mitochondrial genome sequences of the northern spotted owl (*Strix occidentalis caurina*) and the barred owl (*Strix varia*; Aves: Strigiformes: Strigidae) confirm the presence of a duplicated control region

Zachary R. Hanna[1,2,3,4], James B. Henderson[3,4], Anna B. Sellas[4,5], Jérôme Fuchs[3,6], Rauri C.K. Bowie[1,2] and John P. Dumbacher[3,4]

[1] Museum of Vertebrate Zoology, University of California, Berkeley, CA, United States of America
[2] Department of Integrative Biology, University of California, Berkeley, CA, United States of America
[3] Department of Ornithology & Mammalogy, California Academy of Sciences, San Francisco, CA, United States of America
[4] Center for Comparative Genomics, California Academy of Sciences, San Francisco, CA, United States of America
[5] Chan Zuckerberg Biohub, San Francisco, CA, United States of America
[6] UMR 7205 Institut de Systématique, Evolution, Biodiversité, CNRS, MNHN, UPMC, EPHE, Sorbonne Universités, Muséum National d'Histoire Naturelle, Paris, France

Corresponding author
Zachary R. Hanna,
zachanna@berkeley.edu

## ABSTRACT

We report here the successful assembly of the complete mitochondrial genomes of the northern spotted owl (*Strix occidentalis caurina*) and the barred owl (*S. varia*). We utilized sequence data from two sequencing methodologies, Illumina paired-end sequence data with insert lengths ranging from approximately 250 nucleotides (nt) to 9,600 nt and read lengths from 100–375 nt and Sanger-derived sequences. We employed multiple assemblers and alignment methods to generate the final assemblies. The circular genomes of *S. o. caurina* and *S. varia* are comprised of 19,948 nt and 18,975 nt, respectively. Both code for two rRNAs, twenty-two tRNAs, and thirteen polypeptides. They both have duplicated control region sequences with complex repeat structures. We were not able to assemble the control regions solely using Illumina paired-end sequence data. By fully spanning the control regions, Sanger-derived sequences enabled accurate and complete assembly of these mitochondrial genomes. These are the first complete mitochondrial genome sequences of owls (Aves: Strigiformes) possessing duplicated control regions. We searched the nuclear genome of *S. o. caurina* for copies of mitochondrial genes and found at least nine separate stretches of nuclear copies of gene sequences originating in the mitochondrial genome (*Numts*). The *Numts* ranged from 226–19,522 nt in length and included copies of all mitochondrial genes except *tRNA^{Pro}*, *ND6*, and *tRNA^{Glu}*. *Strix occidentalis caurina* and *S. varia* exhibited an average of 10.74% (8.68% uncorrected *p*-distance) divergence across the non-tRNA mitochondrial genes.

## INTRODUCTION

The chicken (*Gallus gallus*) was the first avian species with a complete mitochondrial genome assembly (*Desjardins & Morais, 1990*). Subsequently, researchers assembled the mitochondrial genomes of members of the Paleognathae (e.g., ostriches, emus, kiwis) and other members of the Galloanserae (ducks, chicken-like birds) and recovered the same gene order found in the mitochondrial genome of the chicken, which led to the conclusion that the mitochondrial genome of the chicken is representative of the ancestral avian gene order (*Desjardins & Morais, 1990*; *Mindell, Sorenson & Dimcheff, 1998a*; *Haddrath & Baker, 2001*; *Gibb et al., 2007*). Almost a decade after publication of the chicken mitochondrial genome, *Mindell, Sorenson & Dimcheff (1998a)* described an alternative or, to use their terminology, "novel" avian gene order from that of the chicken, which included a different positioning of *tRNA^{Pro}*, *ND6*, and *tRNA^{Glu}* relative to the control region sequence as well as an additional noncoding segment that they hypothesized was a degraded copy of the control region. A few years later, researchers first described the presence of an intact, duplicated control region in the mitochondrial genomes of *Amazona* parrots (*Eberhard, Wright & Bermingham, 2001*) and the common buzzard *Buteo buteo* (*Haring et al., 2001*).

*Mindell, Sorenson & Dimcheff (1998a)* detected their novel avian gene order in the mitochondrial genomes of taxa in multiple avian orders that spanned a significant portion of Neoaves, but did not detect it in the single owl species that they studied, *Otus asio* (*Mindell, Sorenson & Dimcheff, 1998a*). However, further investigation of owl (Strigiformes) mitochondrial genomes has revealed several surprises.

First, at least three wood owl species (*Strix aluco*, *S. uralensis* (*Brito, 2005*), and *S. varia* (*Barrowclough et al., 2011*)) contain the novel mitochondrial gene order of *Mindell, Sorenson & Dimcheff (1998a)* as well as duplicate control regions. The researchers' use of a primer in *tRNA^{Thr}* to amplify a fragment of the control region suggests that the novel gene order is present in two additional wood owl species, *S. occidentalis* (*Barrowclough, Gutierrez & Groth, 1999*) and *S. nebulosa* (*Hull et al., 2010*). However, the novel gene order was not reported as present in the mitochondrial genome of *S. leptogrammica* (*Liu, Zhou & Gu, 2014*).

Second, some species of eagle-owls (genus *Bubo*) have a large control region (up to ∼3,800 nucleotides) relative to *Strix*, their putative sister genus (*Fuchs et al., 2008*; *Wink et al., 2009*), largely due to a tandem repeat structure in the 3′ end of the control region (*Omote et al., 2013*). Such control region tandem repeat blocks appear to be widespread in Strigidae (*Xiao et al., 2006*; *Omote et al., 2013*). These results suggest that the structures of owl mitochondrial genomes are surprisingly dynamic and in need of further investigation, particularly for species of conservation concern for which portions of the control region are used in population genetic studies (*Barrowclough, Gutierrez & Groth, 1999*; *Haig et al., 2004*; *Hull et al., 2010*; *Hull et al., 2014*).

We here provide the complete mitochondrial genome sequence of both a northern spotted owl (*Strix occidentalis caurina*) and barred owl (*S. varia*). The spotted owl (*S. occidentalis*) is a large and charismatic denizen of dense forests whose range includes the Pacific coast of North America from southwestern British Columbia to southern California and extends eastward into the deserts of the Southwestern United States and southward to central Mexico. The range of the northern spotted owl (*S. o. caurina*) subspecies includes the Pacific Northwest portion of the *S. occidentalis* range from British Columbia south to the Golden Gate strait, California. The US Fish and Wildlife Service has listed *S. o. caurina* as "threatened" under the Endangered Species Act since 1990.

The barred owl (*S. varia*), formerly native east of the Rocky Mountains (*Mazur & James, 2000*), has extended its range into the western US in the last 50–100 years and, from British Columbia to southern California, has become broadly sympatric with the northern spotted owl in the last 50 years. Barred and spotted owls hybridize and successfully backcross (*Haig et al., 2004*; *Kelly & Forsman, 2004*; *Funk et al., 2007*). Mitochondrial DNA sequencing has served as a valuable tool in ascertaining the maternal lineage of western birds, especially in potential hybrids (*Zink, 1994*; *Haig et al., 2004*; *Barrowclough et al., 2005*; *Ruegg, 2008*; *Krosby & Rohwer, 2009*; *Williford et al., 2014*).

Population-level studies of the genetics of *S. occidentalis* and *S. varia* have mainly used two mitochondrial markers, a partial control region sequence (*Barrowclough, Gutierrez & Groth, 1999*; *Haig et al., 2004*; *Barrowclough et al., 2005*) and *cytochrome b* (*cyt b*) (*Haig, Mullins & Forsman, 2004*), although a phylogeographic study of *S. varia* also utilized portions of *ND6* and *COIII* (*Barrowclough et al., 2011*). The sequences of the complete genomes of the mitochondria of these two species will aid researchers in utilizing additional mitochondrial markers in population genetic studies of these owls.

It is well known that mitochondrial genes can transfer to the nuclear genome; such regions of the nuclear genome are sometimes called *Numts* (*Lopez et al., 1994*; *Sorenson & Quinn, 1998*). As a high-quality nuclear genome of *S. o. caurina* is available (*Hanna et al., 2017*), we were able to explore the incidence of *Numts* within the nucleus and investigate which mitochondrial genes have most often transferred. Furthermore, by assessing divergence between mitochondrial genes and their descendent *Numts*, we ascertained the likelihood of them posing problems for phylogenetic and other types of studies.

## METHODS

### *Strix occidentalis* mitochondrial genome assembly

We sourced *Strix occidentalis caurina* DNA from a blood sample collected by a veterinarian from a captive adult female *S. o. caurina* at WildCare rehabilitation facility in San Rafael, California. Found as an abandoned nestling in Larkspur, Marin County, California, WildCare admitted the captive owl as patient card # 849 on 5 June 2005 and named her Sequoia (sample preserved as CAS:ORN:98821; Table 1).

To assemble the *S. o. caurina* mitochondrial genome, we used paired-end Illumina sequence data from nine different genomic libraries constructed, sequenced, and processed

**Table 1 *Strix* specimen data.** We here provide further information regarding the datasets that archive the *Strix* specimens to which we refer throughout the manuscript.

| Specimen | Data publisher | Date accessed | Link to dataset |
|---|---|---|---|
| CAS:ORN:95964 | CAS Ornithology (ORN), California Academy of Sciences, San Francisco, California, United States of America | 2016 Aug 15 | http://ipt.calacademy.org:8080/ipt/resource.do?r=orn |
| CAS:ORN:98821 | CAS Ornithology (ORN), California Academy of Sciences, San Francisco, California, United States of America | 2016 Aug 15 | http://ipt.calacademy.org:8080/ipt/resource.do?r=orn |
| CNHM<USA-OH>:ORNITH:B41533 | Museum of Natural History & Science, Cincinnati Museum Center, Cincinnati, Ohio, United States of America | 2017 Sep 3 | https://www.idigbio.org/portal/records/57d299f0-2cc7-44f7-aa5f-3c2ea175e757 |

as described in *Hanna et al. (2017)*. The raw sequences from sample CAS:ORN:98821 (Table 1) are available from the NCBI Sequence Read Archive (SRA) (SRA run accessions SRR4011595, SRR4011596, SRR4011597, SRR4011614, SRR4011615, SRR4011616, SRR4011617, SRR4011618, SRR4011619, and SRR4011620). For our initial assembly, we used BLATq version 1.02 (*Henderson & Hanna, 2016a*), which was a modification of BLAT version 35 (*Kent, 2002*; *Kent, 2012*), to find Illumina reads that aligned to the *Ninox novaeseelandiae* mitochondrial genome (GenBank Accession AY309457.1) (*Harrison et al., 2004*) (Supplemental Material 1.1.1) and extracted those matching reads using excerptByIds version 1.0.2 (*Henderson & Hanna, 2016b*) (Supplemental Material 1.1.2). We then used SOAPdenovo2 version 2.04 (*Luo et al., 2012*) to assemble those sequences (Supplemental Material 1.1.3).

We used the web version of the NCBI BLAST+ version 2.2.29 tool BLASTN (*Altschul et al., 1990*; *Zhang et al., 2000*; *Morgulis et al., 2008*; *Camacho et al., 2009*) to search the NCBI nucleotide collection (*Johnson et al., 2008*; *Boratyn et al., 2013*; *Benson et al., 2015*; *NCBI Resource Coordinators, 2015*) (NCBI-nt) to assess the completeness of the resulting assembled continuous sequences (contigs) by aligning them to available mitochondrial genome sequences (Supplemental Material 1.1.4). We confirmed that we had assembled a contig with the genes for $tRNA^{Phe}$ through *cyt b* to $tRNA^{Thr}$ that was approximately 18,000 nucleotides (nt) in length, but lacked the complete control region sequence. We used GNU Grep version 2.16 (*Free Software Foundation, 2014*) to search the Illumina reads for matches to the assembled sequence of $tRNA^{Phe}$ or $tRNA^{Thr}$ (Supplemental Material 1.1.5). We found three reads that spanned $tRNA^{Phe}$ and combined them using the Geneious version 9.1.4 *de novo* assembler (*Kearse et al., 2012*; *Biomatters, 2016*) (Supplemental Material 1.1.6). We then extended this assembled contig using a targeted assembly approach with the software PRICE version 1.2 (*Ruby, Bellare & DeRisi, 2013*; *Ruby, 2014*) (Supplemental Material 1.1.7). This PRICE run produced an improved and lengthened assembly after 31 cycles, but the assembly still lacked the complete control region sequence.

We used BLATq version 1.0.2 to align Illumina sequences to the assembly output by PRICE (Supplemental Material 1.1.8) and extracted aligned reads using excerptByIds

**Table 2 Sequence of primers used in Sanger sequencing of control regions.** These are the sequences of all of the primers that we used to amplify control regions 1 and 2 in order to confirm the final sequence of these regions in the mitochondrial genome assemblies.

| Primer name | Relevant region | Species used on | External or internal | Primer sequence (5′ → 3′) | Source |
|---|---|---|---|---|---|
| cytb-F1 | CR1 | *S. o. caurina, S. varia* | External | ATCCTCATTCTCTTCCCCGT | This study |
| 17122R | CR1 | *S. o. caurina, S. varia* | External | GGTGGGGGTTATTATTAACTTT | This study |
| CR1-F1 | CR1 | *S. o. caurina, S. varia* | Internal | CTCSASCAAATCCCAAGTTT | This study |
| CR1-F1-RC | CR1 | *S. o. caurina, S. varia* | Internal | AAACTTGGGATTTGSTSGAG | This study |
| CR1-R2 | CR1 | *S. o. caurina, S. varia* | Internal | GGAGGGCGAGAATAGTTGRT | This study |
| CR1-R2-RC | CR1 | *S. o. caurina, S. varia* | Internal | AYCAACTATTCTCGCCCTCC | This study |
| N1 | CR1 | *S. o. caurina* | Internal | AACATTGGTCTTGTAAACCAA | *Barrowclough, Gutierrez & Groth (1999)* |
| 41R | CR2 | *S. o. caurina* | External | GCATCTTCAGTGCCATGCTT | This study |
| 17572F | CR2 | *S. o. caurina* | External | ATTATCCAAGGTCTGCGGCC | This study |
| 17589F | CR2 | *S. o. caurina* | Internal | GCCTGAAAAACCGCCGTTAA | This study |
| 18327F | CR2 | *S. o. caurina* | Internal | CACTTTTGCGCCTCTGGTTC | This study |
| 19911R | CR2 | *S. o. caurina* | Internal | AGAGAGGCTCTGATTGCTTG | This study |
| ND6-ext1F | CR2 | *S. varia* | External | ACAACCCCATAATAYGGCGA | This study |
| 12S-ext1R | CR2 | *S. varia* | External | GGTAGATGGGCATTTACACT | This study |
| final-CR2F | CR2 | *S. varia* | Internal | TCAAACCAAACGATCGAGAA | This study |
| 18547F | CR2 | *S. varia* | Internal | CTCACGTGAAATCAGCAACC | This study |
| 19088R | CR2 | *S. varia* | Internal | ATTCAACTAAAATTCGTTACAAATCTT | This study |
| 19088R-RC | CR2 | *S. varia* | Internal | AAGATTTGTAACGAATTTTAGTTGAAT | This study |

version 1.0.2 (Supplemental Material 1.1.9). We then performed another PRICE assembly with the same initial contig as before, but with the extracted additional Illumina sequence data (Supplemental Material 1.1.10). This run produced an assembly of one contig of length 18,489 nt after 26 cycles.

We annotated this PRICE assembly using the MITOS WebServer version 605 (*Bernt et al., 2013*) (Supplemental Material 1.1.11), which confirmed that this assembly contained the genes for *tRNA^Phe* through *cyt b* to *tRNA^Thr* followed by control region 1 (CR1), *tRNA^Pro*, ND6, *tRNA^Glu*, and control region 2 (CR2). We searched for repetitive regions using Tandem Repeats Finder version 4.07b (*Benson, 1999*; *Benson, 2012*) (Supplemental Material 1.1.12).

In order to confirm the assemblies of both CR1 and CR2 with longer sequences that could span the repetitive sections of these regions, we designed primers to gene sequences outside of CR1 and CR2 and used Sanger sequencing to obtain verifying sequences across them. We successfully amplified CR2 using a polymerase chain reaction (PCR) with primers 17589F and 41R (Table 2), which primed in *tRNA^Glu* and *tRNA^Phe*, respectively. We then sequenced both ends of the PCR-amplified fragment using BigDye terminator chemistry (Applied Biosystems, Foster City, CA, USA) on an ABI 3130xl automated sequencer (Applied Biosystems, Foster City, CA, USA; Supplemental Material 1.2.1). We also used primer 17572F, which primed in *tRNA^Glu*, and primer 41R (Table 2) to successfully PCR-amplify a slightly longer fragment than above, which also included all of CR2, and then sequenced

across the repetitive section of CR2 using internal primers 18327F and 19911R (Table 2), which primed outside of the repetitive region (Supplemental Material 1.2.2).

We edited the sequences using Geneious version 9.1.4 (*Kearse et al., 2012*; *Biomatters, 2016*) and then used the Geneious mapper to align the sequences to the 19,946 nt preliminary mitochondrial genome assembly (Supplemental Material 1.2.3). These Sanger-derived sequences confirmed that there were nine complete repetitions of a 78 nt motif in CR2 and extended the assembly length to 19,948 nt.

Similarly, we confirmed the CR1 sequence with Sanger-derived sequence data by first PCR-amplifying CR1 with primers cytb-F1 and 17122R (Table 2), which primed in *cyt b* and *ND6*, respectively (Supplemental Material 1.2.4). We visualized the PCR products on a 1% agarose gel, which revealed two PCR products approximately 2,250 and 3,500 nt in length. We re-ran the PCR and gel visualization to confirm this result, which was consistent. We then excised each band from a 1% low melting point agarose gel, performed gel purification using a Zymoclean Gel DNA Recovery Kit (Zymo Research, Irvine, CA, USA), and sequenced the purified fragments using the original external primers as well as the internal primers CR1-F1, CR1-F1-RC, CR1-R2, CR1-R2-RC, and N1 (*Barrowclough, Gutierrez & Groth, 1999*) (Table 2) with BigDye terminator chemistry on an ABI 3130xl automated sequencer. We edited the sequences using Geneious version 9.1.4 and then used the Geneious *de novo* assembler and mapper to assemble the sequences and then align them to the 19,948 nt preliminary mitochondrial genome assembly. We were able to assemble the entirety of the smaller PCR product, but we were unable to completely assemble the CR1 repetitive region in the larger PCR product. Thus, our mitochondrial genome assembly contains the CR1 sequence obtained from the smaller PCR product. The assembly length was then 19,889 nt as the Sanger-confirmed CR1 sequence contained a shorter repetitive region than we assembled with the shorter Illumina sequences. The length of the CR1 repetitive region in the Illumina-sequence-only assembly was also different from the length we expected in the larger PCR product.

In order to use all of the available Illumina sequence data to verify our mitochondrial genome assembly, we took the draft whole genome assembly of *S. o. caurina* (*Hanna et al., 2017*) and replaced scaffold-3674, which was the incomplete assembly of the mitochondrial genome, with the 19,889 nt mitochondrial genome assembly from our targeted assembly methodology (Supplemental Material 1.3.1).

We aligned all filtered Illumina sequences to this new draft reference genome using bwa version 0.7.13-r1126 (*Li, 2013a*) and then merged, sorted, and marked duplicate reads using Picard version 2.2.4 (http://broadinstitute.github.io/picard) (Supplemental Material 1.3.2). We filtered the alignment file to only retain alignments to the preliminary targeted mitochondrial genome assembly using Samtools version 1.3 with HTSlib 1.3.1 (*Li et al., 2009*; *Li et al., 2015*). We then used Samtools and GNU Awk (GAWK) version 4.0.1 (*Free Software Foundation, 2012*) to filter the alignments (Supplemental Material 1.3.3–1.3.4). We next visualized the alignment across the reference sequence in Geneious version 9.1.4 to confirm that the sequence evidence matched our assembly (Supplemental Material 1.3.5).

We annotated the final assembly using the MITOS WebServer version 806 (*Bernt et al., 2013*) (Supplemental Material 1.4.1) followed by manual inspection of the coding loci

and comparison with predicted open reading frames and sequences from *Gallus gallus* (GenBank Accessions NC_001323 (*Desjardins & Morais, 1990*) and AB086102.1 (*Wada et al., 2004*)) in Geneious version 9.1.4. We annotated the repetitive regions using the web version of Tandem Repeats Finder version 4.09 (*Benson, 1999*; *Benson, 2016*) (Supplemental Material 1.4.2). We used bioawk version 1.0 (*Li, 2013b*) and GAWK version 4.0.1 to find goose hairpin sequences in CR1 and CR2 (Supplemental Material 1.4.3). We compared the sequences of the annotated genes in our final mitochondrial genome assembly with those of the incomplete mitochondrial genome assembly that was output as a byproduct of the *S. o. caurina* whole nuclear genome assembly (*Hanna et al., 2017*) in order to evaluate the efficacy of the nuclear genome assembler in assembling mitochondrial genes. We aligned all of the nucleotide sequences of the genes in the final mitochondrial genome against a database of the scaffold-3674 gene nucleotide sequences using NCBI BLAST+ version 2.4.0 tool BLASTN (*Altschul et al., 1990*; *Zhang et al., 2000*; *Morgulis et al., 2008*; *Camacho et al., 2009*) (Supplemental Material 1.4.4).

In order to visualize the binding sites of the primers that we developed to PCR-amplify CR1 and CR2 as well as the primers used by *Barrowclough, Gutierrez & Groth (1999)* to PCR-amplify a portion of CR1 we used Geneious version 9.1.4 (Supplemental Material 1.4.5). We assessed the similarity of CR1 and CR2 by performing a multiple alignment using the Geneious version 9.1.4 implementation of MUSCLE version 3.8.425 (*Edgar, 2004*) (Supplemental Material 1.4.6). In order to assess whether published control region sequences of related species are more similar to CR1 or CR2, we used the web version of NCBI's BLAST+ version 2.5.0 tool BLASTN (*Altschul et al., 1990*; *Zhang et al., 2000*; *Morgulis et al., 2008*; *Camacho et al., 2009*) to search NCBI-nt for sequences similar to CR1 and CR2 (Supplemental Material 1.4.7). As a result of these searches, we aligned the primers used by *Omote et al. (2013)* to PCR-amplify the control region in *Strix uralensis* in their study to our final *S. o. caurina* assembly using Geneious version 9.1.4 (Supplemental Material 1.4.8).

## Nuclear pseudogenation of *Strix occidentalis* mitochondrial genes

In order to examine the incidence of genetic transfer from mitochondria to the nucleus, we examined the draft nuclear genome assembly for evidence of nuclear pseudogenes or nuclear copies of mitochondrial genes (*Numts*) (*Lopez et al., 1994*), in the *S. o. caurina* draft nuclear genome assembly (*Hanna et al., 2017*). We aligned the final *S. o. caurina* mitochondrial genome assembly to the draft nuclear genome assembly using the NCBI BLAST+ version 2.4.0 tool BLASTN (Supplemental Material 1.5.1) using the default threshold Expect value (*E*-value) of 10. We then used GAWK version 4.0.1 to remove all alignments to scaffold-3674, which was the assembly of the mitochondrial genome in the draft nuclear genome assembly. We visually inspected the results to insure that all alignments were of reasonable length and that all *E*-values were <0.0001 (*De Wit et al., 2012*). Indeed, all alignments exceeded 100 nt and all *E*-values were $<1 \times 10^{-25}$. We next used GAWK version 4.0.1 to reformat the BLAST output into a Browser Extensible Data (BED) formatted file (Supplemental Material 1.5.3). In order to determine the mitochondrial genes spanned by each *Numt*, we used BEDTools version 2.26.0 (*Quinlan &*
*Hall, 2010*) to produce a BED file of the intersection of the BED-formatted BLAST output with the BED file output from the MITOS annotation of the final mitochondrial genome assembly (Supplemental Material 1.5.4).

### *Strix varia* mitochondrial genome assembly

In order to assess the divergence between *S. occidentalis* and *S. varia* across all genes of the mitochondrial genome, we constructed a complete *S. varia* mitochondrial genome assembly. We did this by utilizing available whole-genome Illumina data from two *S. varia* individuals collected outside of the zone of contact of *S. varia* and *S. o. caurina* (*Haig et al., 2004*). The main set of *S. varia* whole-genome Illumina data originated from sequencing of a tissue sample collected in Hamilton County, Ohio, United States of America (CNHM<USA-OH>:ORNITH:B41533; Table 1), hereafter "CMCB41533". The paired-end Illumina sequence data was from a genomic library constructed, sequenced, and the data processed as described in *Hanna et al. (2017)*. The raw sequences are available from NCBI (SRA run accessions SRR5428115, SRR5428116, and SRR5428117).

The second *S. varia* individual was from Marion County, Indiana, United States of America (CAS:ORN:95964; Table 1), hereafter "CAS95964". Sequence data from this individual informed the assembly process, but none of these data are included in the final *S. varia* mitochondrial genome assembly (Supplemental Material 1.6.1). The raw sequences are available from NCBI (SRA run accession SRR6026668). We performed adapter and quality trimming of these sequence data using Trimmomatic version 0.30 (*Bolger, Lohse & Usadel, 2014*) (Supplemental Material 1.6.2). For use in only the SOAPdenovo2 assembly, we trimmed the sequences using a different set of parameters and performed error-correction of the sequences using SOAPec version 2.01 (*Luo et al., 2012*) (Supplemental Material 1.6.3).

We constructed the complete *S. varia* mitochondrial genome of sample CMCB41533 by building a succession of assemblies that contributed information to the final assembly from which we extracted the gene sequences. We used partial mitochondrial assemblies of sample CAS95964 to inform the assembly process, but, as we had more sequence data for sample CMCB41533, we chose to only produce a final genome assembly for this sample to compare with that of *S. o. caurina*.

We used two contigs (ContigInput1 and ContigInput2) as the starting material for our final CMCB41533 *S. varia* assembly. In order to generate ContigInput1, we used bwa version 0.7.13-r1126 to align all of the trimmed CMCB41533 paired read 1 and 2 sequences to a reference sequence that included the draft *S. o. caurina* whole nuclear genome along with our final mitochondrial genome assembly (Supplemental Material 1.9.1). We then merged the paired-end and unpaired read alignments, sorted the reads, and marked duplicate reads using Picard version 2.2.4 (Supplemental Material 1.9.2).

We filtered the alignment file to only retain alignments to the final mitochondrial genome assembly using Samtools version 1.3 with HTSlib 1.3.1 (*Li et al., 2009*; *Li et al., 2015*). We then used Samtools and GAWK version 4.0.1 to filter out duplicate reads, low quality alignments, secondary alignments, and alignments where both reads of a pair did not align to the mitochondrial assembly (Supplemental Material 1.9.2–1.9.3). We next visualized

the alignment across the reference sequence in Geneious version 9.1.4 and generated a consensus sequence from the alignment (Supplemental Material 1.9.4). We extracted three sequences from this consensus sequence based on the *S. o. caurina* mitochondrial genome annotations and then used these extracted sequences as three separate seed contigs in an assembly using PRICE version 1.2 (Supplemental Material 1.9.5). This run produced one contig (ContigInput1) of length 9,690 nt after 16 cycles.

The series of assemblies that resulted in ContigInput2, an input to our final *S. varia* assembly, involved first using SOAPdenovo2 version 2.04 to assemble all of the trimmed, error-corrected CAS95964 sequences (Supplemental Material 1.10.1). We extended the output 15,019 nt contig using PRICE version 1.2 (Supplemental Material 1.10.2). After seven cycles, this run produced an assembly of one contig of length 16,652 nt, which included the sequence for $tRNA^{Phe}$ through $tRNA^{Thr}$ and part of CR1. We used this CAS95964 contig to seed a more complete assembly using PRICE version 1.2 with the larger CMCB41533 Illumina sequence dataset (Supplemental Material 1.11.1). After four cycles, this assembly produced one contig (ContigInput2) of length 17,073 nt.

We performed a final assembly using PRICE version 1.2 and the 9,690 nt ContigInput1 and the 17,073 nt ContigInput2 as the initial contigs (Supplemental Material 1.12.1). After two cycles, this assembly produced one contig of length 19,589 nt. We then used Sanger sequencing to confirm the sequences of CR1 and CR2.

We PCR-amplified CR1 with primers cytb-F1 and 17122R (Table 2), which primed in *cyt b* and *ND6*, respectively (Supplemental Material 1.12.2). We then sequenced the fragment using the original external primers as well as the internal primers CR1-F1, CR1-F1-RC, CR1-R2, CR1-R2-RC, and N1 (*Barrowclough, Gutierrez & Groth, 1999*) (Table 2). We PCR-amplified CR2 with primers ND6-ext1F and 12S-ext1R (Table 2), which primed in *ND6* and *12S*, respectively (Supplemental Material 1.12.3). We then sequenced the PCR-amplified fragment using the original external primers as well as the internal primers final-CR2F, 18547F, 19088R, and 19088R-RC (Table 2). We performed all sequencing using BigDye terminator chemistry (Applied Biosystems, Foster City, CA, USA) on an ABI 3130xl automated sequencer (Applied Biosystems, Foster City, CA, USA).

We edited the sequences using Geneious version 9.1.4 and then used the Geneious *de novo* assembler and mapper to assemble the sequences and then align them to the 19,589 nt preliminary mitochondrial genome assembly. These Sanger-derived sequences confirmed that the preliminary assembly was inaccurate in the control regions and reduced the total length to a final size of 18,975 nt. We annotated the assembly using the MITOS WebServer version 605 (Supplemental Material 1.12.4) followed by manual inspection of the coding loci and comparison with predicted open reading frames and sequences from *Gallus gallus* (GenBank Accessions NC_001323 (*Desjardins & Morais, 1990*) and AB086102.1 (*Wada et al., 2004*)) in Geneious version 9.1.4. We annotated the repetitive regions using the web version of Tandem Repeats Finder version 4.09 (*Benson, 1999*; *Benson, 2016*) (Supplemental Material 1.4.2). We used bioawk version 1.0 (*Li, 2013b*) and GAWK version 4.0.1 to find goose hairpin sequences in CR1 and CR2 (Supplemental Material 1.4.3).

## Comparison of *Strix occidentalis* and *Strix varia* mitochondrial genes

In order to compare mitochondrial gene sequences of *S. occidentalis* and *S. varia*, we extracted the nucleotide sequence for all non-tRNA genes (stop codons excluded) from our final *S. o. caurina* and *S. varia* assemblies. We aligned them using MAFFT version 7.305b (*Katoh et al., 2002*; *Katoh & Standley, 2013*; *Katoh, 2016*) (Supplemental Material 1.13.1). We verified the alignments by eye and then used trimAl version 1.4.rev15 (*Capella-Gutiérrez, Silla-Martínez & Gabaldón, 2009*; *Capella-Gutiérrez & Gabaldón, 2013*) to convert the alignments to MEGA format (*Kumar, Tamura & Nei, 1994*; *Kumar, Stecher & Tamura, 2016*) (Supplemental Material 1.13.2). We then used MEGA version 7.0.18 (*Kumar, Stecher & Tamura, 2016*) to calculate the *p*-distance (Supplemental Material 1.13.3) and the corrected pairwise distance (*Tamura & Nei, 1993*) (Supplemental Material 1.13.4) between *S. o. caurina* and *S. varia* for each gene. We calculated a weighted average pairwise distance across all of the genes (Supplemental Material 1.13.5).

## Avian mitochondrial gene order comparisons

We downloaded the mitochondrial genome sequences of *Gallus gallus* (GenBank Accession NC_001323.1) (*Desjardins & Morais, 1990*), *Melopsittacus undulatus* (GenBank Accession NC_009134.1) (*Guan, Xu & Smith, 2016*), *Falco peregrinus* (GenBank Accession NC_000878.1) (*Mindell et al., 1997*; *Mindell, Sorenson & Dimcheff, 1998a*; *Mindell et al., 1999*), *Bubo bubo* (GenBank Accession AB918148.1) (*Hengjiu et al., 2016*), *Ninox novaeseelandiae* (GenBank Accession AY309457.1) (*Harrison et al., 2004*), *Tyto alba* (GenBank Accession EU410491.1) (*Pratt et al., 2009*), *Strix leptogrammica* (GenBank Accession KC953095.1) (*Liu, Zhou & Gu, 2014*), *Glaucidium brodiei* (GenBank Accession KP684122.1) (*Sun et al., 2016*), and *Asio flammeus* (GenBank Accession KP889214.1) (*Zhang et al., 2016*), which were all submitted as complete genomes apart from *Tyto alba*, which was submitted as a partial genome. The *Gallus gallus* mitochondrion represented the ancestral avian order (*Desjardins & Morais, 1990*; *Mindell, Sorenson & Dimcheff, 1998a*; *Haddrath & Baker, 2001*; *Gibb et al., 2007*). The mitochondrial gene order of *Falco peregrinus* was illustrative of the novel gene order first described by *Mindell, Sorenson & Dimcheff (1998a)* with a remnant CR2 (*Gibb et al., 2007*) while the mitochondrial gene order of *Melopsittacus undulatus* exemplified an intact, duplicated control region first described in Psittaciformes by *Eberhard, Wright & Bermingham (2001)*. We visualized the mitochondrial genome sequences and the accompanying annotations using Geneious version 9.1.4. For a coarse assessment of gene similarity, we next used the Geneious version 9.1.4 implementation of MUSCLE version 3.8.425 in order to align all of the owl (Aves: Strigiformes) mitochondrial genomes as well as to align the *S. leptogrammica* mitochondrial genome with our *S. o. caurina* and *S. varia* assemblies.

## RESULTS

We deposited the complete mitochondrial genome sequences of *Strix occidentalis caurina* sample CAS:ORN:98821 and *Strix varia* sample CNHM<USA-OH>:ORNITH: B41533 as NCBI GenBank Accessions MF431746 and MF431745, respectively. The lengths of the final *S. o. caurina* and *S. varia* mitochondrial genome assemblies were 19,889 nt and 18,975 nt,

**Figure 1  Ancestral avian mitochondrial gene order surrounding the control region compared with that of *Strix occidentalis caurina* and *Strix varia*.** The Chicken panel displays the gene order of *Gallus gallus*, which is the presumed ancestral avian gene order. The Spotted Owl panel depicts the gene order of *Strix occidentalis caurina* and the Barred Owl panel depicts the gene order of *Strix varia*. All rRNAs, tRNAs, and protein-coding genes outside of the displayed region exhibit the same order in all of these mitochondrial genomes. "CR" denotes the control region with "CR1" and "CR2" referring to control regions 1 and 2, respectively. We added 100 nucleotides to each of the tRNAs to improve visualization. Apart from the tRNAs, the annotations are to scale relative to each other with the numbers at the top of the figure denoting nucleotides. The orders of the genes outside of the region depicted in this figure are the same in the chicken, spotted owl, and barred owl.

respectively. As for all typical avian mitochondrial genomes, they are circular and code for two rRNAs, 22 tRNAs, and 13 polypeptides (Figs. S1 and S2). The annotations produced by MITOS identified a 1 nt gap that split *ND3*, which is consistent with the untranslated nucleotide and translational frameshift seen in *ND3* in some other bird species (*Mindell, Sorenson & Dimcheff, 1998b*), including owls (Strigiformes) (*Fuchs et al., 2008*).

Both the *S. o. caurina* and *S. varia* mitochondrial genomes contain a duplicated control region (Fig. 1). In both genomes, CR1 and CR2 each include a C-rich sequence near the 5′ end, the goose hairpin (*Quinn & Wilson, 1993*), which is identical across the two species and across CR1 and CR2. The *S. o. caurina* CR1 contains a 70 nt motif repeated 6.8 times near the 3′ end while CR2 includes two sets of tandem repeats near the 3′ end of the region, a 70 nt motif repeated four times followed by 9.5 repetitions of a 78 nt motif (Table 3).

The *S. o. caurina* CR1 and CR2 share a conserved central block of 1,222 nt with only two mismatches between CR1 and CR2 (Fig. 2). This conserved block includes 202 nt of the 5′ portion of the repetitive regions. The *S. varia* CR1 and CR2 share a conserved 1,041 nt central sequence stretch containing five mismatches. In CR1, this conserved block begins in the 3′ 57 nt of the CR1 repetitive region, but in CR2 it does not extend into the repetitive region. The 5′and 3′ regions surrounding the conserved central blocks of the control regions in both *S. o. caurina* and *S. varia* are more divergent from each other.

We obtained an alignment (88.37% identity) of 1,429 nt from the 5′ ends of the *S. o. caurina* and *S. varia* CR1 sequences, but it included fifteen gaps (Fig. 3). In contrast, the more 3′ repetitive sections of the *S. o. caurina* and *S. varia* CR1 sequences yielded an uninformative alignment with numerous, long gap regions. Similarly to CR1, the 5′ ends of the *S. o. caurina* and *S. varia* CR2 sequences aligned well (90.62% identity), yielding a 1,300 nt alignment that included four gaps. However, the alignment of the 3′ region of the CR2 sequences was uninformative with numerous, long gaps due to conflicts between the 78 nt motif repetitive regions of the two CR2 sequences. We found no evidence of mitochondrial pseudogenes in the control region sequences of either *S. o. caurina* or *S. varia*.

Across all of the 35 genes that were present in the previous, incomplete *S. o. caurina* assembly that was produced as a byproduct of the assembly of the *S. o. caurina* whole
**Table 3 Tandem repeat annotations.** This summarizes the repetitive regions of the northern spotted owl (*Strix occidentalis caurina* or *S. o. caurina*) and barred owl (*S. varia*) mitochondrial genomes annotated by Tandem Repeats Finder. "Period size" refers to the size of the repeated motif. "Copy number" refers to the number of copies of the repeat in the region. "Consensus size" is the length of the consensus sequence summarizing all copies of the repeat, which may or may not be different from the period size. "Percent matches" refers to the percentage of nucleotides that match between adjacent copies of the repeat. "Percent indels" refers to the percentage of indels between adjacent copies of the repeat. We present the percent composition of each of the four nucleotides in the repetitive region. We have included the genomic regions that intersect each repetitive span in the "Region" column. "CR1" and "CR2" refer to control region 1 and control region 2, respectively.

| Taxon | Coordinates (nt) | Region | Period size (nt) | Copy number | Consensus size (nt) | Percent matches (%) | Percent indels (%) | A (%) | C (%) | G (%) | T (%) |
|---|---|---|---|---|---|---|---|---|---|---|---|
| *S. o. caurina* | 10,267–10,309 | *ND4* | 18 | 2.3 | 19 | 84 | 4 | 25 | 46 | 0 | 27 |
| *S. o. caurina* | 15,066–15,162 | CR1 | 22 | 4.3 | 22 | 70 | 7 | 37 | 27 | 6 | 28 |
| *S. o. caurina* | 15,169–15,311 | CR1 | 67 | 2.1 | 67 | 83 | 8 | 40 | 31 | 6 | 21 |
| *S. o. caurina* | 16,243–16,715 | CR1 | 70 | 6.8 | 70 | 98 | 1 | 39 | 21 | 4 | 33 |
| *S. o. caurina* | 16,245-16,715 | CR1 | 139 | 3.4 | 139 | 99 | 0 | 39 | 22 | 4 | 33 |
| *S. o. caurina* | 16,403–16,515 | CR1 | 37 | 3.2 | 37 | 61 | 27 | 40 | 23 | 3 | 32 |
| *S. o. caurina* | 17,679–17,795 | CR2 | 44 | 2.6 | 45 | 87 | 4 | 43 | 30 | 4 | 21 |
| *S. o. caurina* | 17,719–17,795 | CR2 | 22 | 3.5 | 22 | 89 | 0 | 45 | 31 | 3 | 19 |
| *S. o. caurina* | 18,798–19,076 | CR2 | 70 | 4.0 | 70 | 99 | 0 | 39 | 21 | 4 | 34 |
| *S. o. caurina* | 18,800–19,076 | CR2 | 139 | 2.0 | 139 | 100 | 0 | 39 | 21 | 4 | 34 |
| *S. o. caurina* | 18,958-19,070 | CR2 | 37 | 3.2 | 37 | 61 | 27 | 40 | 23 | 3 | 32 |
| *S. o. caurina* | 19,110–19,853 | CR2 | 78 | 9.5 | 78 | 99 | 0 | 41 | 15 | 15 | 27 |
| *S. varia* | 15,126–15,209 | CR1 | 22 | 3.8 | 22 | 82 | 4 | 36 | 27 | 4 | 30 |
| *S. varia* | 15,193–15,340 | CR1 | 67 | 2.2 | 68 | 83 | 1 | 37 | 32 | 8 | 22 |
| *S. varia* | 17,384–17,482 | CR2 | 22 | 4.4 | 23 | 87 | 5 | 41 | 34 | 5 | 19 |
| *S. varia* | 18,548–18,951 | CR2 | 78 | 5.2 | 77 | 93 | 2 | 40 | 17 | 15 | 26 |

nuclear genome (*Hanna et al., 2017*), we only found one mismatch with our complete assembly, which occurred between the two *ND1* sequences. This assembly improves upon the previous version by providing the complete sequences of *ND6*, *tRNA^Pro^*, and the two control regions.

The *S. o. caurina* CR1 is 2,021 nt in length and the *S. varia* CR1 is 1,686 nt long. In both species, the 5′ end of CR1 borders *tRNA^Thr^* and the 3′ end is adjacent to *tRNA^Pro^*, then *ND6*, and then *tRNA^Glu^* (Fig. 1). The initial 1,104 nt of the *S. o. caurina* CR1 are identical to a *S. o. caurina* partial control region sequence (GenBank Accession AY833630.1) (*Barrowclough et al., 2005*). All of the top 100 matches of the BLASTN searches of the *S. o. caurina* CR1 to NCBI-nt were to either *S. occidentalis* or *S. varia* control region sequences deposited by other researchers, as we expected.

CR2 follows *tRNA^Glu^* and is 2,319 nt in length in *S. o. caurina* and 1,719 nt long in *S. varia*. The initial 549 nt of the *S. o. caurina* CR2 matches the beginning of the D-loop sequence of an annotated complete genome of a *Bubo bubo* mitochondrion (GenBank Accession AB918148.1) (*Hengjiu et al., 2016*). One of the top 100 matches of the BLASTN searches of the *S. o. caurina* CR2 to NCBI-nt, which had the highest total score (2,177) and query coverage (96% versus 36–41% for the other matches) of the top 100 matches, was to a *S. uralensis* control region sequence (GenBank Accession AB743794.1)

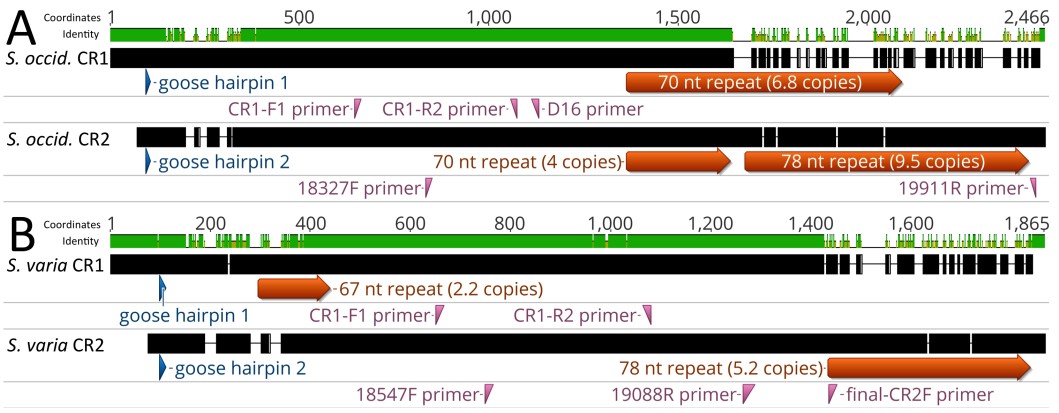

**Figure 2** **Alignment of control regions 1 and 2 within *Strix occidentalis caurina* and *Strix varia*.** (A) depicts an alignment of the *Strix occidentalis caurina* control regions 1 and 2. (B) displays an alignment of the *Strix varia* control regions 1 and 2. The numerical coordinates at the top of each panel correspond to the coordinates of the alignment. Black rectangles for each control region denote continuous sequence, whereas intervening horizontal lines denote gaps in the alignment. The sequence identity rectangle is green at full height when there is agreement between the sequences, yellow at less than full height when the sequences disagree, and flat in gap regions. The location of the goose hairpin sequence in each control region is annotated in blue. The alignment locations of the primers we developed to amplify control regions 1 and 2 as well as the D16 primer used by *Barrowclough, Gutierrez & Groth (1999)* to amplify a portion of control region 1 are annotated in reddish purple.

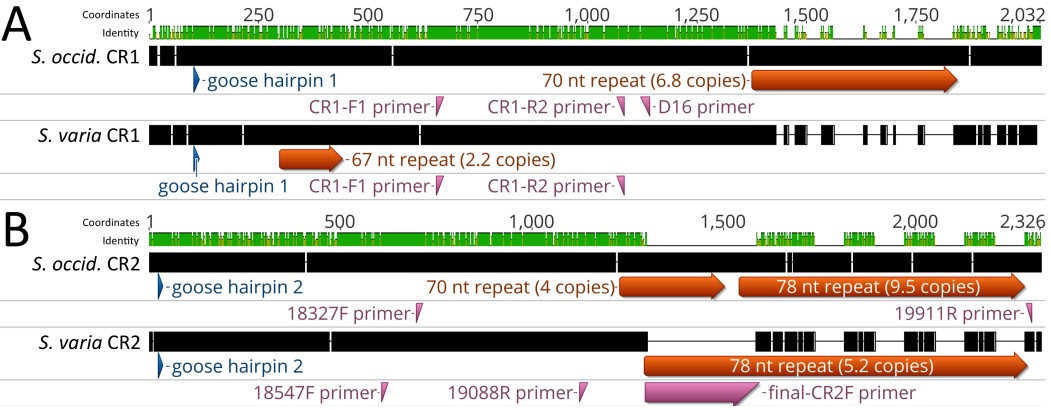

**Figure 3** **Alignment of *Strix occidentalis caurina* control regions 1 and 2 with those of *Strix varia*.** (A) depicts an alignment of the *Strix occidentalis caurina* control region 1 with that of *Strix varia*. (B) displays an alignment of the *Strix occidentalis caurina* control region 2 with that of *Strix varia*. The numerical coordinates at the top of each panel correspond to the coordinates of the alignment. Black rectangles for each control region denote continuous sequence, whereas intervening horizontal lines denote gaps in the alignment. The sequence identity rectangle is green at full height when there is agreement between the sequences, yellow at less than full height when the sequences disagree, and flat in gap regions. The location of the goose hairpin sequence in each control region is annotated in blue. The alignment locations of the primers we developed to amplify control regions 1 and 2 as well as the D16 primer used by *Barrowclough, Gutierrez & Groth (1999)* to amplify a portion of control region 1 are annotated in reddish purple. The annotation of primer final-CR2F is elongated as it is situated across a gap region in the alignment.
(*Omote et al., 2013*). The majority of the primers used by *Omote et al. (2013)* to PCR-amplify the control region in *S. uralensis* align within and around the *S. o. caurina* CR2. Four of the control-region-specific primers align to the middle of CR2 in our *S. o. caurina* sequence, which is identical to the middle of the *S. o. caurina* CR1 sequence. Perhaps most crucially, the primer L16728 aligns in the forward direction in *tRNA*$^{Glu}$ such that it would amplify CR2, if present in the species.

As we mentioned in the methodology, our PCR-amplification of the *S. o. caurina* CR1 using primers that spanned from *cyt b* to *ND6* yielded two products approximately 2,250 and 3,500 nt in length (Fig. S3). The sequences of these fragments were identical in the *cyt b* and *ND6* portions as well as in the adjacent CR1 sections except when they entered the repetitive region at the 3′ end of CR1. We were only able to obtain sequence spanning the entirety of this repetitive region in the 2,250 nt fragment. This was largely due to the fact that the 3,500 nt fragment, in addition to the 70 nt motif repetitive section observed in the sequence of the 2,250 nt fragment, contained another repetitive region on the *tRNA*$^{Pro}$ side of the 70 nt motif region with at least 13.1 copies of a 67 nt motif. We did not find any copies of *tRNA*$^{Pro}$ or *ND6* in the *S. o. caurina* nuclear genome, but we did find nuclear copies of *cyt b* and *tRNA*$^{Thr}$. With *ND6* absent from the nuclear genome, PCR-amplification using primers in *cyt b* and *ND6* should only generate mitochondrial genome fragments. Additionally, the *cyt b* and *tRNA*$^{Thr}$ sequence in both the 2,250 nt and 3,500 nt fragments did not match the nuclear genome copies of these genes. In summary, we believe that both of these fragments were mitochondrial in origin and this evidence suggests that at least two different versions of the mitochondrial genome were present in this *S. o. caurina* individual.

The annotations of the mitochondrial genome sequences of the owls (Aves: Strigiformes) *Tyto alba*, *Ninox novaeseelandiae*, *Strix leptogrammica*, *Glaucidium brodiei*, and *Asio flammeus* indicate that those owls all share the same mitochondrial gene order as *Gallus gallus*, the ancestral avian mitochondrial gene order (*Desjardins & Morais, 1990*; *Mindell, Sorenson & Dimcheff, 1998a*; *Haddrath & Baker, 2001*) (Fig. 1). Our alignment of the *S. leptogrammica* mitochondrial genome to the mitochondrial genomes of other owls, including our *S. o. caurina* and *S. varia* assemblies, resulted in a poor, gap-filled alignment of the genes from the second half of the *S. leptogrammica cyt b* sequence through *ND6* to *tRNA*$^{Phe}$. We could not obtain a reasonable alignment of the last 210 nt of the *S. leptogrammica* D-loop adjacent to the *tRNA*$^{Phe}$ sequence to our *S. o. caurina* and *S. varia* assemblies or to the mitochondrial genomes of any of the other Strigiformes whose sequences we examined. Additionally, alignment of the *S. leptogrammica* mitochondrial genome with our *S. o. caurina* and *S. varia* assemblies yielded an *ND5* alignment with seven gaps and numerous mismatches (85.60% and 84.82% identity to *S. o. caurina* and *S. varia*, respectively). Together, these results suggest that the *S. leptogrammica* sequence potentially contains significant errors in the sequences of the genes from *ND5* through *ND6* to *tRNA*$^{Phe}$.

We found 29,520 nt of *Numt* sequences in the draft *S. o. caurina* nuclear genome assembly spanning nine *Numts* (Table 4). The *Numts* ranged in length from 226–19,522 nt and had an average length of 3,280 nt. The *Numts* provided evidence of nuclear copies of all

Hanna et al. (2017), *PeerJ*, DOI 10.7717/peerj.3901

**Table 4  Mitochondrion-derived nuclear pseudogenes (*Numts*) identified in the *Strix occidentalis caurina* nuclear genome sequence and statistics of the results of BLASTN searches.** We indicate the mitochondrial genes that a *Numt* spans in the "Genes included" column. If a *Numt* spans more than two genes, we indicate the first and last genes that it spans as well as a gene in the middle of the *Numt* in order to indicate the direction that the *Numt* extends. The *Numt* additionally spans all of the intervening genes in such cases. "Start mtDNA" and "End mtDNA" indicate the mitochondrial genome assembly sequence positions and "Start Scaffold" and "End Scaffold" denote the nuclear genome assembly contig/scaffold sequence positions in the alignments of the mitochondrial genome assembly to the nuclear genome assembly. "% ID" indicates the percentage of identical matches in an alignment. "*E*-value" is the Expect value. "Bit score" is a log-scaled version of the alignment score. We characterized some of the *Numts* by examining more than one alignment and concluding that a *Numt* spanned across those individual alignments.

| *Numt* # | Genes included | Start mtDNA | End mtDNA | Nuclear genome scaffold | Start scaffold | End scaffold | Orientation | % ID | *E*-value | Bit score | Length alignment (nt) | Length *numt* (nt) |
|---|---|---|---|---|---|---|---|---|---|---|---|---|
| 1 | *tRNA^Phe* - 12S - 16S | 1 | 2,225 | scaffold478 | 47,666 | 49,858 | + | 79.92 | 0.0 | 1,565 | 2,261 | 19,522 |
| | 16S | 2,367 | 2,645 | scaffold478 | 49,871 | 50,143 | + | 87.81 | 2.16e−84 | 322 | 279 | – |
| | 16S - ND2 - *tRNA^Asn* | 2,706 | 5,223 | scaffold478 | 50,161 | 52,680 | + | 80.66 | 0.0 | 1,921 | 2,549 | – |
| | *tRNA^Asn* - COI - *tRNA^Ser*2 | 5,219 | 6,932 | scaffold478 | 57,635 | 59,328 | + | 83.22 | 0.0 | 1,552 | 1,716 | – |
| | *tRNA^Ser*2 - *tRNA^Asp* - COII | 6,988 | 7,103 | scaffold478 | 59,382 | 59,496 | + | 87.18 | 1.41e−26 | 130 | 117 | – |
| | ATP6 - ND4 - ND5 | 8,382 | 13,249 | scaffold478 | 59,498 | 64,306 | + | 80.59 | 0.0 | 3,672 | 4,893 | – |
| | cyt b | 14,047 | 14,733 | scaffold478 | 44,785 | 45,459 | + | 82.82 | 1.92e−169 | 604 | 687 | – |
| | *tRNA^Thr* | 14,729 | 14,878 | scaffold478 | 46,066 | 46,222 | + | 82.80 | 1.09e−27 | 134 | 157 | – |
| 2 | 16S | 1,682 | 2,603 | scaffold215 | 5,517,239 | 5,518,161 | − | 81.97 | 0.0 | 773 | 932 | 923 |
| 3 | *tRNA^Ser*2 - ATP8 - ND3_a | 6,989 | 9,584 | scaffold215 | 5,513,222 | 5,515,749 | − | 79.01 | 0.0 | 1,690 | 2,615 | 2,528 |
| 4 | 16S - *tRNA^Leu*2 | 2,290 | 2,788 | scaffold632 | 1,548,886 | 1,549,372 | + | 77.50 | 6.14e−70 | 274 | 511 | 487 |
| 5 | ND1 - *tRNA^Gln* - ND2 | 2,810 | 4,646 | scaffold167 | 11,322,764 | 11,324,590 | + | 80.54 | 0.0 | 1,400 | 1,840 | 2,732 |
| | ND2 - *tRNA^Asn* - COI | 4,692 | 5,597 | scaffold167 | 11,324,598 | 11,325,495 | + | 83.68 | 0.0 | 846 | 907 | – |
| 6 | *tRNA^Glu* - ND2 - COI | 3,851 | 5,526 | scaffold1500 | 35,914 | 37,582 | − | 84.21 | 0.0 | 1,620 | 1,678 | 1,669 |
| 7 | ND2 - *tRNA^Asn* - *tRNA^Tyr* | 4,500 | 5,348 | scaffold173 | 750,945 | 751,785 | − | 81.40 | 0.0 | 680 | 855 | 841 |
| 8 | ND5 | 12,082 | 12,310 | scaffold143 | 586,822 | 587,047 | + | 81.30 | 3.83e−42 | 182 | 230 | 226 |
| 9 | CR1 | 15,026 | 15,640 | scaffold294 | 2,356,468 | 2,357,059 | − | 83.07 | 9.17e−148 | 532 | 620 | 592 |
| | CR2 | 17,677 | 18,195 | scaffold294 | 2,356,468 | 2,356,986 | − | 80.87 | 9.70e−108 | 399 | 528 | – |

**Table 5  Divergence of *Strix occidentalis caurina* and *Strix varia* at all protein-coding genes.** This provides the number of base substitutions per site for all mitochondrial protein-coding genes and rRNAs between the mitochondrial sequences of *Strix occidentalis occidentalis* and *S. varia*. *P*-distance refers to an uncorrected pairwise distance while TN93 refers to the pairwise distance corrected by the Tamura-Nei 1993 model (*Tamura & Nei, 1993*).

| Gene | Number of sites in alignment (nt) | *p*-distance | Distance with TN93 model |
|---|---|---|---|
| *12S* | 984 | 5.79% | 6.61% |
| *16S* | 1,589 | 5.48% | 6.14% |
| *ATP6* | 681 | 9.10% | 11.07% |
| *ATP8* | 165 | 14.55% | 20.81% |
| *COI* | 1,548 | 7.88% | 9.31% |
| *COII* | 681 | 9.10% | 11.23% |
| *COIII* | 783 | 7.54% | 8.89% |
| *cyt_b* | 1,140 | 9.21% | 11.35% |
| *ND1* | 957 | 10.66% | 13.46% |
| *ND2* | 1,038 | 9.34% | 11.60% |
| *ND3_a* | 174 | 10.92% | 13.96% |
| *ND3_b* | 174 | 11.49% | 14.86% |
| *ND4* | 1,377 | 10.31% | 13.12% |
| *ND4L* | 294 | 11.22% | 14.29% |
| *ND5* | 1,818 | 9.19% | 11.29% |
| *ND6* | 516 | 9.69% | 14.71% |

mitochondrial genes, except $tRNA^{Pro}$, *ND6*, and $tRNA^{Glu}$, the three genes between CR1 and CR2. *Numt* #9 (Table 4) aligns to both CR1 and CR2 with the alignments extending into the conserved block shared by the control regions. The portion of genome scaffold-294 aligned to CR2 for this *Numt* is 519 nt, whereas the length aligned to CR1 is 592 nt. As we could not be sure of which control region was incorporated into the nuclear genome, we have provided information for both alignments and derived the length of the *Numt* from the alignment to CR1 (Table 4).

   *Strix occidentalis caurina* and *S. varia* display an average of 10.74% (8.68% uncorrected p-distance) base substitutions per site across the 2 rRNA genes and 13 polypeptide genes (the non-tRNA mitochondrial genes) (Table 5). The lowest number of base substitutions per site occurs within *16S* and the highest within *ATP8* (Table 5).

## DISCUSSION

Sequences of most mitochondrial genes can often be recovered from high-throughput short-read sequencing data if genome complexity is not too great. Algorithms using short-read data have more difficulty assembling low-complexity or repetitive regions due to an inability to span these regions. Thus, assembly of complete mitochondrial genome sequences can be more difficult when such genomes include regions of low-complexity. The sequence of the avian control region can both contain blocks of tandem repeats (*Omote et al., 2013*) and be duplicated (*Eberhard, Wright & Bermingham, 2001*; *Haring et al., 2001*). Moreover, the presence of multiple controls regions that are similar or identical, which has

been observed in snakes (*Kumazawa et al., 1996*), can cause problems with assembly. In such situations, additional types of sequencing data that complement short-read data may be necessary in order to obtain an accurate and complete assembly of the mitochondrial genome. This proved to be the case in our study where the longer Sanger-derived sequence data were crucial in obtaining the complete sequence of the lengthy, repeat-rich control regions in *S. o. caurina* and *S. varia*. Although *Brito (2005)* and *Barrowclough et al. (2011)* inferred the presence of a duplicated control region structure in the mitochondrial genomes of at least three wood owl species, *Strix aluco*, *S. uralensis*, and *S. varia*, they did not sequence complete mitochondrial genomes. They likely deduced that a duplication was present from the appearance of multiple bands on agarose gels resulting from PCR-amplification of portions of the mitochondrial control region. Here we describe the first complete genome sequences of the mitochondrion of an owl (Aves: Strigiformes) with a duplicate control region.

The mitochondrial genomes of *S. o. caurina* and *S. varia* exhibit the novel avian gene order first described by *Mindell, Sorenson & Dimcheff (1998a)* for several bird orders, but not reported by them as present in the owl *Otus asio*. As mentioned above, this duplicated control region structure and novel gene order has previously been reported in the mitochondrial genome of *S. varia* (*Barrowclough et al., 2011*) and the congeners *S. aluco* and *S. uralensis* (*Brito, 2005*). The novel gene order was previously implied for *S. occidentalis* by the placement of primer N1 in *tRNA^{Thr}* by *Barrowclough, Gutierrez & Groth (1999)* to PCR-amplify the control region (CR1) fragment used in their study. *Hull et al. (2010)* also used the *Barrowclough, Gutierrez & Groth (1999)* N1 primer to PCR-amplify the control region in their study of *S. nebulosa*, so we can infer that the *S. nebulosa* mitochondrion also possesses the *Mindell, Sorenson & Dimcheff (1998a)* novel gene order. Notably, this mitochondrial gene order was not reported as present in *S. leptogrammica* (*Liu, Zhou & Gu, 2014*). However, our alignments of this mitochondrial genome to our *S. o. caurina* and *S. varia* sequences as well as the sequences of other owl mitochondrial genomes indicated problems with the *S. leptogrammica* sequence from *cyt b* through *ND6* to *tRNA^{Phe}*. If we then leave aside the *S. leptogrammica* sequence, available evidence suggests that the novel gene order and duplicate control region structure is present across the genus *Strix*.

The primers developed by *Barrowclough, Gutierrez & Groth (1999)* to PCR-amplify a fragment of the control region (CR1) in *S. occidentalis* have been used extensively in additional genetic studies of owl species (*Haig et al., 2004*; *Brito, 2005*; *Marthinsen et al., 2009*; *Hull et al., 2010*; *Barrowclough et al., 2011*; *Hausknecht et al., 2014*). The *Barrowclough, Gutierrez & Groth (1999)* control region primers D16 (the most 3′ of their primers) and D20 (more 5′ relative to primer D16) align to a region conserved between CR1 and CR2, although the length of the distance from the binding site of primer N1 in *tRNA^{Thr}* to the CR2 sites of primers D16 and D20 (3,742 nt and 3,392 nt, respectively, in our *S. o. caurina* assembly) likely reduces the probability of this second primer binding site causing problems in the PCR-amplification of the CR1 fragment.

The second control region appears intact, not degraded as found in some other avian taxa (*Mindell, Sorenson & Dimcheff, 1998a*). This gene order corresponds to the "Type D Duplicate CR gene order" of *Gibb et al. (2007)* and the "Duplicate CR gene order I"

of *Eberhard & Wright (2016)*. The goose-hairpin structure is typically found near the beginning of the control region in avian mitochondria (*Marshall & Baker, 1997*; *Randi & Lucchini, 1998*; *Bensch & Härlid, 2000*) and, in agreement with what we found, it appears in the beginning of the intact, duplicated control region sequences in the genomes of *Amazona* (*Eberhard, Wright & Bermingham, 2001*) and additional parrot mitochondria (*Eberhard & Wright, 2016*).

The lengths of the *S. o. caurina* CR1 and CR2 (2,021 nt and 2,319 nt, respectively) and of the *S. varia* CR1 and CR2 (1,686 nt and 1,719 nt , respectively) are all shorter than the length reported for the control regions of some species in the sister genus of owls, *Bubo* (*Wink et al., 2009*), which have lengths up to approximately 3,800 nt due to tandem repeats in the 3′ end of the control region (*Omote et al., 2013*). Similar tandem repeat blocks occur in the control regions of several other owl species in the family Strigidae (*Xiao et al., 2006*; *Omote et al., 2013*). The length of the tandem repeat motif unit is 78 nt in the 3′ end of the control region sequences of *Bubo blakistoni*, *Bubo virginianus*, *Strix uralensis* (*Omote et al., 2013*), and *Strix aluco* (*Xiao et al., 2006*); 78 nt is also the length of the motif in the longest tandem repeat block in both the *S. o. caurina* and *S. varia* CR2 (Table 3).

As we previously mentioned, both *S. uralensis* and *S. aluco* exhibit a duplicated control region structure in their mitochondrial genomes (*Brito, 2005*). Neither *Omote et al. (2013)* nor *Xiao et al. (2006)* report the presence of a duplicated control region structure in either *S. uralensis* or *S. aluco*, respectively, in their discussions of the repetitive content of the control region sequences of these two species. It is not overtly clear from their methodologies which control region they sequenced. The precise primer combinations used for the PCR-amplification and sequencing of the control region of the *Bubo* species and *S. uralensis* are not provided by *Omote et al. (2013)*, but mapping the primer sequences used by the researchers to our *S. o. caurina* genome suggests that, if the structure of the *S. uralensis* mitochondrial genome shares that of *S. o. caurina*, they likely sequenced CR2 in at least *S. uralensis* and in the *Bubo* species if a CR2 was present. We are unsure how placement of primers in *cyt b* and *12S*, as reported in the methodology of *Xiao et al. (2006)* could PCR-amplify a single control region sequence for *S. aluco*, given the duplicated control region structure (*Brito, 2005*).

The duplicated control region structure is unreported in Strigiformes outside of *Strix*, but we infer that it is also likely present in *Bubo* due to the positioning of primers used for PCR-amplification of the control region in previous studies (*Marthinsen et al., 2009*; *Omote et al., 2013*). If also present in *Bubo*, then the duplicate control region structure appears to have arisen in the common ancestor of *Strix* and *Bubo*, but a proper phylogenetic test of this hypothesis with increased taxon sampling is warranted. Further work on the structure of control region sequences in *Strix* and related taxa is needed to elucidate the pattern of evolution of this region across the Strigidae phylogeny.

Although inconclusive and warranting further investigation, our evidence for two versions of the 3′ repetitive region of CR1 suggests that mitochondrial heteroplasmy is present in this *S. o. caurina* individual. Mitochondrial heteroplasmy due to tandem repeat variability in the control region has been shown to occur in other bird species (*Berg, Moum & Johansen, 1995*; *Mundy, Winchell & Woodruff, 1996*). Previous work has suggested that

the most likely mechanism by which the gain and loss of such tandem repeat elements occurs in the mitochondrial control region is that the repetitive region forms a stable, single-stranded secondary structure and there is slippage during replication (*Levinson & Gutman, 1987*; *Wilkinson & Chapman, 1991*; *Fumagalli et al., 1996*; *Faber & Stepien, 1998*). Greater numbers of repeats may improve the stability of the secondary structure (*Faber & Stepien, 1998*). Utilizing sequence from the 3′ region of CR1 for population genetic study of *S. o. caurina* is not likely to be useful due to the variability (in terms of the number of copies of the tandem repeat motifs in this region) that is potentially present within a single individual.

The 29,520 nt of *Numt* sequence in the draft *S. o. caurina* nuclear genome assembly is more than triple the 8,869 nt of *Numt* sequence found in a *Gallus gallus* draft nuclear genome assembly (*Pereira & Baker, 2004*). The 3,280 nt average *Numt* size exceeds the average size in all of the eukaryotic genomes examined by *Richly & Leister (2004)*. There are markedly fewer control region *Numts* in the *S. o. caurina* draft genome assembly than found in a *Gallus gallus* draft genome assembly (*Pereira & Baker, 2004*). We only found one control region *Numt* (Table 4). Indeed the longest *Numt*, *Numt* #1, extends through almost the entire mitochondrial genome sequence including from $tRNA^{Phe}$ through $tRNA^{Thr}$, immediately adjacent to, but ending at CR1. The percentage identity of the nuclear pseudogenes with the true mitochondrial genes ranges from 77.5–87.81%, so care must be taken to insure that *Numts* are not PCR-amplified in place of mitochondrial gene sequences. As the control region is the mitochondrial segment that has been used most often in studies of the population genetics of *Strix* species (*Barrowclough, Gutierrez & Groth, 1999*; *Haig et al., 2004*; *Barrowclough et al., 2005*; *Brito, 2005*; *Hull et al., 2010*; *Barrowclough et al., 2011*), it is encouraging that only one, short *Numt* included CR1 or CR2 (Table 4). As long as researchers PCR-amplify sequences that span beyond the 592 nt *Numt* #9, they should have confidence in amplifying the true mitochondrial control regions.

The average pairwise sequence divergence between *S. occidentalis* and *S. varia* has been previously reported as 13.9% for a 524 nt section of CR1 (*Haig et al., 2004*). This exceeds the weighted average of 10.74% (8.68% uncorrected p-distance) that we calculated across the non-tRNA mitochondrial genes (Table 5), which is unsurprising as the control region is known to be rapidly evolving in birds (*Quinn & Wilson, 1993*). However, the pairwise sequence divergence between *S. occidentalis* and *S. varia* appears higher in *ND3*, *ND4L*, *ND6*, and *ATP8* (Table 5) than in the 524 nt CR1 portion. *Hull et al. (2010)* found uncorrected *p*-distances of 13.73–13.93% for *ND2* and 14.58–14.81% for the control region between *S. nebulosa* and *S. occidentalis*. *Wink & Heidrich (1999)* calculated uncorrected *p*-distances of 8.15–11.72% for *cyt b* sequence pairwise comparisons of six *Strix* species (*S. aluco*, *S. butleri*, *S. nebulosa*, *S. rufipes*, *S. uralensis*, and *S. woodfordii*). Of these six *Strix* species, only *S. aluco* x *S. uralensis* are known to hybridize (*McCarthy, 2006*), albeit in captivity (*Scherzinger, 1982*), and their *cyt b* pairwise divergence was 8.15%, the lowest of those calculated by *Wink & Heidrich (1999)* between *Strix* species. Our *cyt b* uncorrected *p*-distance value between *S. o. caurin a* and *S. varia* was 9.21%, which is also on the lower end of the range of the *Wink & Heidrich (1999)* *Strix* interspecific divergences. Overall, however, high levels of interspecific pairwise divergence of mitochondrial DNA seem to be

typical for the genus *Strix*, even for species able to hybridize. We anticipate that these whole mitochondrial genome resources will be useful to those with an interest in developing new mitochondrial markers to study the genetics of *S. o. caurina*, *S. varia*, and related taxa.

## ACKNOWLEDGEMENTS

We thank WildCare, San Rafael for graciously providing blood samples from Sequoia. We thank the Cincinnati Museum Center for providing a barred owl (*Strix varia*) tissue sample. We thank Laura Wilkinson for assistance with laboratory work. We generated genetic sequence data at the Center for Comparative Genomics, California Academy of Sciences.

### Funding

Funds provided by Michael and Katalina Simon (to John P. Dumbacher); the Louise Kellogg Fund, Museum of Vertebrate Zoology, University of California, Berkeley (to Zachary R. Hanna); and the National Science Foundation Graduate Research Fellowship (DGE 1106400 to Zachary R. Hanna) made this work possible. Any opinion, findings, and conclusions or recommendations expressed in this material are those of the authors and do not necessarily reflect the views of the National Science Foundation. This work used data produced by the Vincent J. Coates Genomics Sequencing Laboratory at the University of California, Berkeley, supported by NIH S10 Instrumentation Grants (S10 RR029668, S10 RR027303). The funders had no role in study design, data collection and analysis, decision to publish, or preparation of the manuscript.

### Grant Disclosures

The following grant information was disclosed by the authors:
Louise Kellogg Fund, Museum of Vertebrate Zoology, University of California, Berkeley.
National Science Foundation: DGE 1106400.
National Institutes of Health: S10 RR029668, S10 RR027303.

### Competing Interests

The authors declare there are no competing interests.

### Author Contributions

- Zachary R. Hanna conceived and designed the experiments, performed the experiments, analyzed the data, wrote the paper, prepared figures and/or tables, reviewed drafts of the paper.
- James B. Henderson, Anna B. Sellas and John P. Dumbacher conceived and designed the experiments, performed the experiments, analyzed the data, contributed reagents/materials/analysis tools, wrote the paper, prepared figures and/or tables, reviewed drafts of the paper.
- Jérôme Fuchs conceived and designed the experiments, performed the experiments, analyzed the data, wrote the paper, reviewed drafts of the paper.

- Rauri C.K. Bowie conceived and designed the experiments, contributed reagents/materials/analysis tools, wrote the paper, reviewed drafts of the paper.

### DNA Deposition

The following information was supplied regarding the deposition of DNA sequences:

Complete mitochondrial genome sequence of *Strix occidentalis caurina* CAS:ORN:98821 deposited as NCBI GenBank Accession MF431746.

Complete mitochondrial genome sequence of *Strix varia* CNHM<USA-OH>:ORNITH: B41533 deposited as NCBI GenBank Accession MF431745. The raw sequences for *Strix varia* sample CAS:ORN:95964 are available from NCBI (SRA run accession SRR6026668).

### Data Availability

BLATq deposited at Zenodo DOI: 10.5281/zenodo.61136 available at https://github.com/calacademy-research/blatq.

excerptByIDs deposited at Zenodo DOI: 10.5281/zenodo.61134 and available at https://github.com/calacademy-research/excerptByIDs.

### Supplemental Information

Supplemental information for this article can be found online at http://dx.doi.org/10.7717/peerj.3901#supplemental-information.

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
