# Peer review of "Complete mitochondrial genome sequences of the northern spotted owl (Strix occidentalis caurina) and the barred owl (Strix varia; Aves: Strigiformes: Strigidae) confirm the presence of a duplicated control region"

_PeerJ, doi:10.7717/peerj.3901_

## Round 0.1 · original submission · Minor Revisions

Dear authors
this is a well written ms, but it needs some finetuning as you can see from the comments of the reviewers.

Regards
Michael Wink
Academic editor

Reviewer 1 ·

Basic reporting

gel picture really needed?

Experimental design

some details could be in an appendix

Validity of the findings

no comment

Additional comments

when repeatly mentioning a software product or mito-genome, no need to cite again
numbers in tables should be right-justified
line 101: able to explore

Reviewer 2 ·

Basic reporting

In general, I found the manuscript to be well written; even some of the denser parts of the Methods section were clearly worded and easy to follow. The figures are of high quality, but some could be cut (or combined, or moved to Supplementary Materials) without significantly impacting the manuscript; see detailed comments below. Raw sequences have been deposited in NCBI’s Sequence Read Archive, but I was unable to find Genbank accession numbers in the text for the two new sequences produced for Strix o. caurina and Strix varia; these should be provided.

Experimental design

The research described in this manuscript is original and adds to the growing body of work on avian control region evolution. Extensive detail is provided in the main text on the methods used, with further details given in the Supplementary Materials. If anything, the Methods text could be trimmed, moving more of the nitty-gritty to Supplementary Materials.

Validity of the findings

Overall, the authors make good use of their data to address questions outlined in the Introduction. Employment of both next-gen and Sanger sequencing techniques was critical to obtaining accurate mt genome sequences for these owls. In fact, the authors could further emphasize the fact that their study represents a cautionary case in which a next-gen-based assembly resulted in an incorrect mt genome sequence. Given the prevalence of next-gen sequencing, combined with the fact that control region duplications have been found in a growing number of disparate avian taxa, it is important to consider the possibility of control region duplications whenever sequencing avian mt genomes.

Additional comments

I would suggest re-thinking the sub-headings used within the Methods section. Since you have separate descriptions regarding the sequencing/assembly of the two species’ genomes, it would be nice if each description were labeled as such, and the two descriptions were structured (divided) into consistent sub-sections.

My detailed comments are listed below by line number.
l.36 At the end of the abstract, you might want to add a sentence regarding the shortcomings of short-read sequencing/assembly methods in cases like yours.
l.42 “which led to”
l.43 “of the chicken is representative”
l.100 Technically, you can’t cite (Hanna et al. 2017) in this way, since it is not published. More correct: (Hanna et al. unpublished) or (Hanna et al. in prep.)
l.160 “amplify a slightly longer”
l.184 Did the length of the longer PCR product jive with the (incorrect) Illumina-based assembly?
l.209-214 The purpose of this comparison was not clear to me.
l.227-256 This section could be moved, so that it is presented after the description of the S.varia assembly.
l.227-228 The alligator sequence comparison isn’t mentioned/discussed later on in the text, so it’s not clear why it’s included here. The alligator sequence is included in Figure 6, but as noted in the figure comment below, it could be cut without detracting from the conclusions.
l.238-240 See previous comment.
l.247-248 See above.
l.256 “Nuclear pseudogenation” might be a better term here, since you are referring to pseudogenes that appear in the nuclear genome. There are also pseudogenes of mt genes that appear within rearranged mt genomes (e.g., see Eberhard & Wright 2016).
l.258 See previous comment.
Related comment: Is there any evidence of mitochondrial pseudogenes, such as those found in some parrots (see Eberhard & Wright 2016)? There seems to be a short stretch of sequence on the 5’ side of CR1, before the hairpin sequence; any chance that it’s similar to tRNA-Glu or ND6?
l.287-292 It might be better to present the primary sequence source first, and then describe the secondary one.
l.299 It would be helpful to briefly explain that the two contigs are.
l.421 Is it possible to discount the hypothesis that the long fragment was a numt? (or, to really play devil’s advocate, is it possible to determine that one or both of the fragments are indeed mitochondrial?)
l.446-447 This is interesting (though I don’t have any idea why this might be)!
l.453 As someone who works with other (non-owl) birds, I found these divergences to be quite high for taxa that can hybridize. Are these levels of divergence typical for owls?
l.461-462 Identical or nearly identical CRs (e.g., as observed in snakes, e.g., Kumazawa et al. 1996, MBE 13:1242-1254) could also cause assembly problems.
l.466 “complete and accurate assembly”
l.469 “inferred the presence”
l.471 “but rather deduced that a duplication was present from the appearance”
l.475 “The mitochondrial genomes…”
l.502 “as found in some other avian taxa”
l.508 Not just Amazona parrots; also found in several other parrot taxa with independently evolved CR duplications (see Eberhard & Wright 2016)
l.517 “a proper phylogenetic test of this hypothesis, with increased taxon sampling, is warranted”
Table 1 At least one of the primers listed in the table was designed by other researchers (e.g., N1, Barrowclough et al. 1999), and this should be noted. It would be good to add a column for references, or add footnotes, to cite sources for primer sequences, and also note which were designed by the authors.
Table 2 Some of the information presented in the table (e.g., Alignment score, Entropy) are not discussed in the text, and don’t seem crucial to the results/discussion.
Table 3 Per earlier comment (l.256), “Nuclear pseudogenes (Numts)…” Two of the columns, E-value and Bit score, provide information that is never discussed in the text, so could be eliminated. If they are not cut, then Bit score needs to be defined in the caption.
Figures 1&2 Both of these are beautifully drawn, but are largely redundant; could be replaced with a single figure that shows gene order surrounding the CRs.
Figures 3&4 The figure caption should also state that the locations of the goose hairpin and primer binding sites are shown. To streamline the figure a bit, the yellow boxes labeled “control region 1” and “control region 2” could be cut, since the regions are already labeled on the left.
Figure 5 The description given in the text is probably enough, so it’s not necessary to include this figure. If desired, it could be moved to Supplementary Information.
Figure 6 The caption refers to panels B & C, but no such labels appear in the figure. Although the information presented in the figure is not incorrect, it illustrates only a subset of the gene orders that have been thus far documented for birds; the purpose of illustrating the owl gene order within an incomplete context is not clear. I would recommend a figure that illustrates the gene order around the owl CRs (as suggested in lieu of Figs. 1&2), and then cite (and discuss, if you wish) other papers that have provided more complete discussion of mt gene order rearrangements in birds (e.g., Mindell et al. 1998, and more recently, Gibb et al. 2007 and Eberhard & Wright 2016).

---

## Round 0.2 · accepted · Accept

Congratulations. The revision is ok by now and is accepted.
Thanks for publishing in PeerJ

Greetings

Michael Wink
Academic editor